# Patients' decision-making, experiences and preferences regarding pixantrone treatment in relapsed or refractory diffuse large B-cell lymphoma: study protocol for a longitudinal mixed methods study

Lothar E van Hoogdalem,[1] Claire Siemes,[2] Pieternella J Lugtenburg,[2] Jan J V Busschbach,[1] Sohal Y Ismail[1]

[1]Department of Psychiatry, Section of Medical Psychology & Psychotherapy, Erasmus MC, Rotterdam, The Netherlands
[2]Department of Hematology, Erasmus MC, Rotterdam, The Netherlands

**Correspondence to**
Dr Lothar E van Hoogdalem;
l.vanhoogdalem@erasmusmc.nl

## ABSTRACT

**Introduction** There is a lot of speculation about why and how patients decide to use invasive treatment in an advanced stage of cancer, but the body of research is limited. The present longitudinal qualitative and quantitative study reflects real-life practice of pixantrone use and aims to collect data on patients' considerations for, expectations of and experiences with pixantrone and trajectories in their quality-of-life (QoL) values in a Dutch clinical setting. Hence, two questions emerge. Why do patients choose for this treatment, while the treatment success rate is limited and curation cannot be achieved? And second, once chosen, what conditions would patients like to satisfy and how do they experience the treatment?

**Methods and analysis** This is a non-interventional longitudinal and multicentre study. Patients are eligible if they are >18 years, have never been treated with pixantrone before, have an Eastern Cooperative Oncology Group performance score ≤2, have a relapsed or refractory diffuse large B-cell lymphoma and have been treated with at least two prior regimens. The decision to treat patients with pixantrone has been taken by the treating physician before patients are asked to participate in the study. If patients refuse study participation after being informed by the investigator, reasons for refusal (if given) will be recorded. Participants will receive at least three interviews accompanied by three QOL questionnaires. Based on the required sample size, we aim to include 20 patients over a period of 2 years.

**Ethics and dissemination** The Medical Ethical Committee of Erasmus MC, Rotterdam, The Netherlands, has approved this study. The results will be disseminated in peer-reviewed journals and major international conferences. The study is non-interventional and falls therefore not under Medical Research Involving Human Subjects Act (In Dutch: Wet medisch-wetenschappelijk onderzoek met mensen; WMO). Hence, this study is approved to be carried out in the Erasmus MC. Each other participating centre will receive this approval and will separately undergo the ethical approval to be able to participate. In addition to the ethical approval, the

### Strengths and limitations of this study

► The use of mixed methods for data collection.
► The present research reflects real-life practice of pixantrone use.
► Results will provide knowledge about patients' reasoning in an advanced stage of cancer.
► Gaining a better understanding of treatment-related decision-making and treatment experiences could facilitate decision-making for future patients and doctors.

participating centres need to obtain written informed consent of their patients. Given the non-interventional nature of this study, a study registration was considered but deemed unnecessary. The study will be conducted in accordance with the Declaration of Helsinki (Tokyo, Venice, Hong Kong and Somerset West amendments). A sequential identification number will be automatically attributed to each patient that has given consent to participate in the study. This number will identify the patient and must be included on all documents. Only the main researcher can link the code to the patient's identity.

## INTRODUCTION

There is a lot of speculation about why and how patients decide to use invasive treatment in an advanced stage of cancer, but the body of research is limited.[1–3] In an advanced and incurable stage, patients and their family have to cope with difficult decisions on further treatment.[4] A little increase in survival time must be weighed against the intensity of the treatment. A meaningful task of the healthcare system is to support decision-making for both physicians and patients. Experiences of other patients can be thought to be helpful in that respect.[5] The present longitudinal

mixed methods study reflects real-life practice of pixantrone use in patients with relapsed or refractory diffuse large B-cell lymphoma (DLBCL) and aims to collect data on the patients' considerations for, expectations of and experiences with pixantrone and trajectories in their quality-of-life (QoL) values in a Dutch clinical setting.

In 2007, 1572 patients were diagnosed with aggressive non-Hodgkin's lymphoma (NHL) in the Netherlands and it is expected that the incidence will increase to almost 1900 patients in the year 2020 due to the ageing of the population.[6] After NHL diagnosis and during its treatments, patients' health-related quality of life (HRQOL) can be affected in a negative way.[7] For instance, NHL survivors reported significant worse psychological and physical HRQOL.[8] New developments in treatment practices are therefore essential. Treatment with pixantrone (Pixuvri, Servier) is a new development, which is indicated as monotherapy for the treatment of adult patients with DLBCL who have been previously treated with two or more regimens. Pixantrone, administered intravenously in the hospital, was developed to reduce the anthracycline-associated cardiotoxicity while retaining efficacy. The drug, a novel cytotoxic aza-anthracenedione, has been proven to be efficacious in patients with relapsed or refractory aggressive B-cell lymphoma.[9 10] Patients treated with pixantrone demonstrated an overall response rate of 37% compared with an overall response rate of 14% for patients treated with comparator agents. The progression-free survival for patients treated with pixantrone was 2.1 months longer compared with patients treated with comparator agents. Besides pixantrone, there is no approved substitute available. None of the drugs currently being evaluated show adequate potential as single agent in the treatment of patients with relapsed DLBCL ineligible for autologous stem cell transplant (ASCT). The drug is available on the market since fall 2014 and just recently included in the national guidelines for patients who undergo a third-line or fourth-line treatment for DLBCL.[11] Therefore, in the Netherlands, a very small number of patients have so far been treated with this drug. Hence, no data are thus far reported regarding patients' decision-making and experiences with this treatment. Collecting and reporting of such data are the main objective of this study.

Treatment decision-making involves the evaluation of all available information, weighing the risks and benefits, personal preferences and eventually the selection of the best alternative. Information is one of the most important preconditions for participation in decision-making.[12] From a systematic review about information giving and decision-making in patients with advanced cancer,[1] it is known that the need for information is often underestimated by the doctors. Moreover, informed decisions on further treatment cannot be made without knowing or understanding the actual prognosis. Adequate information giving reduces anxiety, creates realistic expectations and gives a sense of control. Thus, it contributes to an increase in HRQOL. Patients' preferences for being involved in decision-making differ from their desire for information, almost 66 per cent of patients in an end-of-life situation would like to be involved to a certain extent.[6 13] But this may vary during the course of the disease: patients who become more ill usually want to pass control to the doctor,[14] while those who have improved after the first treatment often want more involvement in subsequent decisions.[6] Other relevant determinants are age, gender, socioeconomic status and education. Currently, the age at diagnosis for NHL is approximately 66 years.[15] On average, older patients with cancer prefer to receive less information about their disease and treatment and prefer a less active role in making treatment decisions. They are also less likely to collect and analyse all relevant information in order to make an optimal decision.[16] Older patients in Europe have, on average, a lower educational level than younger ones and therefore may be less confident in the results of treatments and less aware of medical innovations.[17] Older people may also think that decision-making is the physicians' responsibility.[18] When it comes to differences in gender, it seems that men prefer to play a less active role in decision-making than women.[19] Patients from lower social classes are often disadvantaged because of the physicians' misperception of their need for information and their ability to take part in the healthcare process.[20] Another precondition for decision-making is patients' competence.[21] Competence contains the patient's ability to understand that their choices have consequences. A competent person has the capacity to reach a reasonable decision by comprehending all relevant information and by weighing the benefits and the risks. Patients' decision-making abilities may be negatively affected by fears, anxiety and difficulties in understanding the given information.[22] Additionally, opinions of family members are also relevant.[23] No data are yet available on differences between ethnic populations in comparable disease settings as patients for whom pixantrone is indicated on the target parameters of this study (decision-making, preferences and experiences). Therefore, we include both Western and non-Western patients and do not expect ethnic differences on the parameters under research. There also might be a difference between patients that start pixantrone treatment in a relative early disease setting (third-line treatment) versus those that start the treatment in a later disease setting (fourth line and later). Patients that start the treatment with pixantrone in late disease settings have inherently undergone more treatments which may make their decisions, experiences and preferences regarding pixantrone treatment different compared with patients in relatively earlier disease settings.

Given the expected increase in NHL incidence rates due to the increasing average age and the speculations about why and how these patients decide to accept invasive treatment, there is still much to be done to understand patients' reasoning in this advanced stage of cancer. The present research reflects real-life practice of pixantrone use and aims to collect data on the patients'

considerations for, expectations of and experiences with pixantrone and trajectories in their QoL values in a Dutch clinical setting.

## Objectives

The primary objective of this non-interventional study is to qualitatively explore patients' decision-making, experiences and preferences regarding pixantrone treatment for their DLBCL. Hence, two questions emerge. Why do patients choose for this treatment, while the treatment success rate is limited and curation cannot be achieved? And second, once chosen, how do they experience the treatment? The secondary objective of this study is to assess changes in QOL among patients during their treatment for DLBCL with pixantrone. Since the QOL drops down over the course of the therapy, the expected QOL trajectory will be in a negative direction. Therefore, the changes in QOL values will be used to compare (1) patients that respond to treatment with pixantrone versus those that do not respond to treatment, (2) patients that start the drug treatment with pixantrone in third-line treatment versus those that start the treatment in the fourth line and later and (3) patients that show complete response to pixantrone within patient QOL value differences in the disease progression-free period versus the QOL value at the moment of disease progression.

## METHODS AND ANALYSIS
### Patient and public involvement

Patients or public were not involved. Patient participation will be sought for two phases: (1) conducting research: agenda setting for patient and public involvement meetings in collaboration with the research team and providing a governance function to ensure that researchers are acting responsibly; and (2) sharing and using research knowledge: sharing knowledge and learning about the research to other relevant stakeholders (eg, by giving verbal updates, writing reports, presenting at or attending conferences and meetings) and guiding the direction of future research.

### Study population

Eligible patients are adults (>18 years) with a current diagnosis of relapsed or refractory DLBCL that have been treated with at least two prior lines of therapy, have never have been treated with pixantrone before and have an Eastern Cooperative Oncology Group performance status of 0–2.[24] Patients who are participating in an (experimental) drug trial other than pixantrone or those with an uncontrolled medical disease that could compromise patients' safety will not be included. Although patients using pixantrone are scarce, we will strive to include at least 20 patients over a period of 2 years.

### Design and procedure

The purposive sample of 20 patients will be recruited from a wide range of tertiary care hospitals in The Netherlands

due to the scarcity of the research population. Currently, the study is approved to be carried out in four medical centres. The decision to treat patients with pixantrone has been taken by the treating physician before patients are asked to participate in the study. The physician of the participating medical centre will inform the researcher when they are going to treat a patient with pixantrone. If a patient is willing to participate, the first author (LvH, junior researcher, MSc Neuropsychology, man) will invite him or her by telephone for the first interview. Participation is completely voluntary: subjects can leave the study at any time for any reason if they wish to do so without stating any reason and any consequences for their medical treatment (dropout). If they have a reason that they want to tell, it will be recorded. The participants are made aware of this in the patient information letter they receive before signing informed consent. All data collected up to the moment of dropout will be stored. If premature termination has taken place for reasons other than patient-initiated factors (eg, medical reasons, sudden death), a final measurement (interview +questionnaires) will take place, if the participant has not withdrawn consent for that reason. In case of a dropout, we will seek further to include more patients, within the possibilities of our time schedule. Those patients that participate will receive at least three interviews accompanied by three QOL questionnaires (see also figure 1). Measurement point T2 will only consist of the three questionnaires. If a patient does not master the Dutch language, the use of a translated questionnaire or an interpreter for interviews will be considered and offered to the patient. Both the interview techniques as the questionnaires are according to the standard clinical practice. The administration of both can take either place in the hospital or at the patient's home. The choice of the study venue will be left to the patient, whichever is more comfortable for him or her. The patient may be accompanied by their spouse, another family member or caregiver during the interview. Those are classified as observers, and their input will not be used in data analysis. It is conceivable that patients become upset during the interview. Alongside the interviewer (SI, senior researcher, PhD Medical Science, man), a junior researcher (LvH) will be present. Both are well-trained psychologists and are instructed how to cope with such situation. During the interview, this junior researcher will ensure correct recording of the interview and will be taking paper and pencil notes (backup data). Both the interviewer and the junior researcher have adequate knowledge of qualitative methods based on previous research projects and education. The researchers have no prior relationship with the participants and all participants will be provided with information about the study and the research team before starting the interviews. The interviews will approximately last 1 hour and an additional 15 min to complete the QOL questionnaires. Data collection is scheduled as follows (see also figure 1):

► T0=0–8 days prior to pixantrone treatment cycle 1: interview and questionnaires.

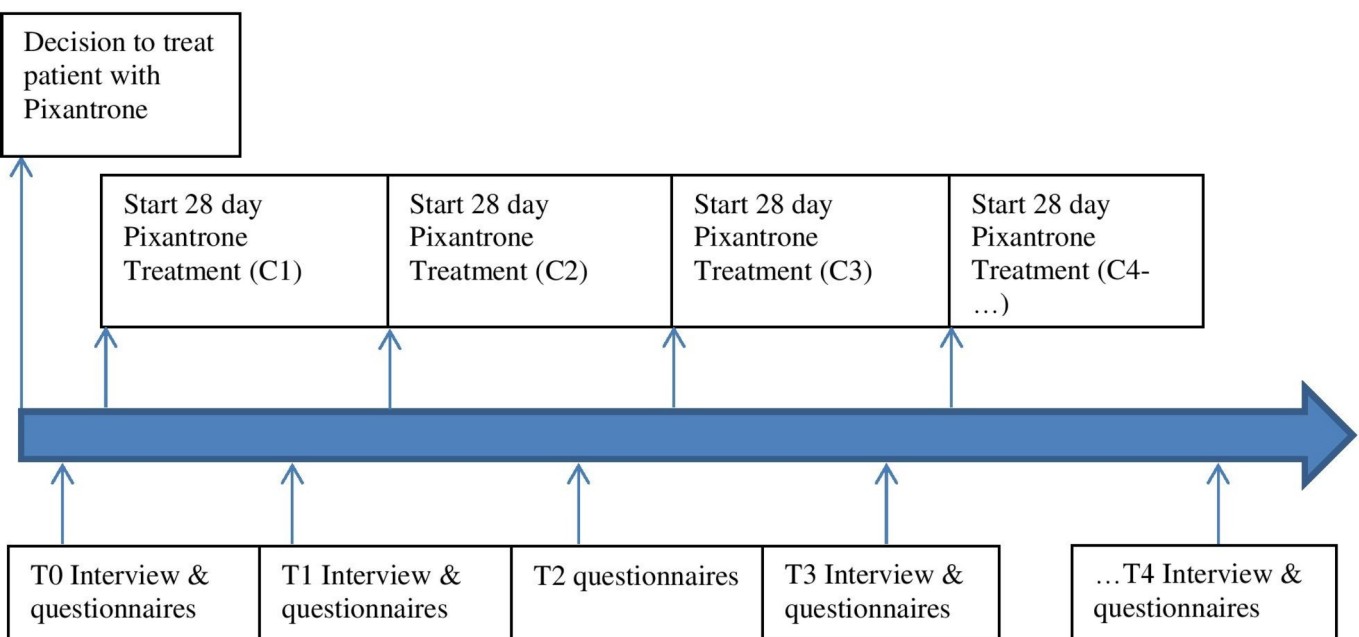

**Figure 1** Patients' flow and measurement moments in the study.

- T1=3–4 weeks, between weeks 3 and 4 during pixantrone treatment cycle 1 (C1): interview and questionnaires.
- T2=7–8 weeks, between weeks 3 and 4 during pixantrone treatment cycle 2 (C2): questionnaires only.
- T3=11–12 weeks, between weeks 3 and 4 during pixantrone treatment cycle 3 (C3): interview and questionnaires.

In case of disease progression or treatment discontinuation during C1–C3: the interview at T3 will be the last interview. For patients that show or remain in complete response, partial response or stable disease after C3 (approximately 20%–25%), the closing interview including questionnaires is planned at:

- T4=within week 1–2 after discontinuation of pixantrone due to disease progression or initiated by and at the discretion of the physician or the patient.

### Data collection
#### Qualitative interviews
Data from the interviews will be analysed according to the principles of grounded theory,[25 26] which is aimed at the construction of theory grounded in the data from which it is developed. A more detailed description of this approach is explained in the next paragraph. This approach is chosen because of the lack of knowledge about patients' decision-making, experiences and preferences regarding pixantrone treatment. To explore patients' perspectives, semistructured interviews will be conducted. Interviews will be taped and transcribed verbatim and if necessary translated into Dutch. Tape recordings will be altered to remove identifying information. Interviews will be anonymised (person, place names, staff, family members). We do not plan to return the transcripts to participants for

comments or corrections. There is no evidence that use of certain checks improves research quality where the primary purpose of the research is theory development.[27] The interview contains four major topics: *decision-making-related factors* (eg, weighing pros and cons, the involvement of others), *treatment-related factors* (eg, expectations about benefits and side effects), *physician-related factors* (eg, communication, trust in physician) and *patient-related factors* (eg, disease knowledge, treatment history). An additional file shows the topic list in more detail (see online supplementary additional file 1). This topic list is developed by the research team with reference to their previous projects and through discussion of the objectives and study aim.

The transcripts will first be coded using open coding. Next, these codes will be grouped into concepts after which the concepts will be piled into categories that cover all relevant information. To maximise efficiency and reliability compared with paper and pencil analysis, the qualitative software programme NVivo (V.11) will be used for analysis. First, we will generate a grid according to Miles and Huberman's method in order to be able to compare the data between the different time points.[28] At least two independent researchers (LvH and SI) will read through the transcripts twice while listening to the tape recordings and organise them into a table. Words or phrases will be combined together in order to generate a covering category. This process goes on until the two researchers separately worked through the whole transcript and data saturation has been reached. The two researchers then will jointly cluster the derived categories into themes. Thereby, they will identify the underlying uniformities of the categories and further sharpen the conceptual structure of each theme. Finally, within each theme, responses

van Hoogdalem LE, *et al. BMJ Open* 2019;9:e026505. doi:10.1136/bmjopen-2018-026505

will be evaluated across the different subgroups to search for similarities and differences. Reporting of qualitative findings will adhere to the consolidated criteria for reporting qualitative research (COREQ).[29]

### Cancer Quality of Life Questionnaire-Core 30

The European Organization for Research and Treatment of Cancer Quality of Life Questionnaire (EORTC QLQ-C30) is a core generic questionnaire to measure the quality of life developed for use in clinical cancer trials.[30] This questionnaire will be administered at T0–T4 (see also figure 1) to measure the trajectory of the QoL of these patients while being treated with pixantrone. The EORTC QLQ-C30 is a 30-item questionnaire, with a global health status score, five functional scales and three symptom scales. The questionnaire is documented as having both good validity and reliability.[30] The EORTC QLQ-C30 questionnaire and the manual are free of charge to academic users and Dutch translations are available. The EORTC QLQ-C30 will be obtained in Dutch, English, Arabic, Hindi and Turkish, due to the possible multicultural character of the study population.

### Quality of Life Questionnaire-High Grade 29

The EORTC QLQ-NHL-LG20 (low-grade NHL) and the EORTC QLQ-NHL-HG29 (intermediate-grade and high-grade NHL) are recently developed NHL-specific questionnaires.[31] The development of these questionnaires (LG20 and HG29) are in its final (fourth) validation phase. The EORTC-NHL-HG29 will be administered to patients in this study (1) to gain some additional insight into disease-specific experiences and (2) to, perhaps, contribute to further validation of these additional disease-specific items. This questionnaire concerns NHL and treatment-related symptoms, including emotional impact, physical condition, symptom burden, worries, health and functioning and neuropathy. Items are scored on a 4-point Likert scale. A higher score reflects more or worse symptoms.[31] This questionnaire will be administered at T0–T4 (see also figure 1).

### EuroQol 5 Dimensions 5 Levels

The EuroQol 5-Dimension 5-Level (EQ-5D-5L) instrument was developed by a multidisciplinary group of researchers from seven centres across five countries.[32] It has become one of the most widely used generic measures of health in Europe and has become commonly used in economic evaluation. The validity and reliability are generally supported.[33] The questionnaire consists of two parts. The first part is a five-dimensional questionnaire. The five dimensions are mobility, self-care, usual activities, pain/discomfort and anxiety/depression. They each have five levels and together define 243 health states. The second part is a Visual Analogue Scale which represents the patients' judgments of his or her own health state on a scale from 0 to 100 (100 being the best possible health state). Patients are classified into the EQ-5D by self-completion or interviewer administration. The EQ-5D will be administered at T0–T4 (see also figure 1).

### Sample size calculation

We aim to gather a sufficient number of participants to reach data saturation during analysis. Previous interview studies on patients' decision-making and perspectives on healthcare have shown that a study population of 20 participants should be sufficient in order to reach such saturation.[34 35] After each subject, the interview will be transcribed as soon as possible and via this way monitored if new concepts arise after each patient (test for data saturation). Hence, data saturation is reached when no new concepts/information arises. The number of 20 is an approximation of the number obtained from literature that is required to reach data saturation.[34 35] In that case, there is enough information to replicate the study, and further coding does not bring out new information. Additionally, we will increase the sample size if needed to achieve data saturation. If different themes arise between treatment line, age or gender, it will be discussed in the discussion section.

### Statistical analysis

Participants will be asked to fill out the, EQ-5D and the recently developed EORTC QLQ-30 with additional 29 disease-specific items (QLQ HG-29). The questionnaires will be interpreted in accordance with the manual (reference data) after a linear transformation into a scale with a range of 0–100. A higher score refers to a higher stage of QOL. To analyse patients' QoL trajectories (ie, how many shows an increase, decrease and no change) over time, Reliable Change Indexes (RCIs) will be calculated. Using the RCI, one can determine whether an individual change score on a measure is large enough that it is unlikely that this change is the consequence of measurement error and can, therefore, be considered as a 'real change.' We will calculate the RCIs with the RCI formula described by Jacobson and Truax.[36] Difference scores on the QOL questionnaire will be calculated between baseline and subsequent measures. RCI analyses cannot take into account variation in time. This limitation should not be a problem if protocol adherence is in place. Subsequently, patients will be divided into three categories: if a difference score (follow-up score—baseline score) was smaller than the RCI, the person was assigned to the 'no change' category; if a difference score was positive and greater than the RCI, the person was assigned to the 'increase' category; and if a difference score was negative and greater than the RCI, the person was assigned to the 'decrease' category. Finally, if available/obtained, data from a normative population will be compared with patients' data using Mann-Whitney tests to examine whether RCI changes differ between the two groups.[37 38] For all analyses, we will use the Statistical Package for the Social Sciences V.21.0 (SPSS) and a p value less than 0.05 will be considered statistically significant.

## DISCUSSION

We presented a protocol for studying patients' decision-making, experiences and preferences regarding pixantrone treatment in heavily pretreated patients with relapsed or refractory DLBCL. Oncological treatment decision-making has become increasingly difficult for patients and their physicians over the past few years.[39] The quantity, complexity and adverse effects of various therapies make it difficult to assess costs and benefits of different treatment options. Therefore, it is important for the physicians to know the patients' experiences and preferences regarding decision-making and the chosen treatment. Patient participation in decision-making is recommended because it increases control of patients with cancer over their well-being and results in more patient-oriented decisions[40] and may lead to improved health outcomes.[41 42] Results of this study will contribute to improving the understanding of (1) the decision-making process and once chosen (2) the experiences of an additional treatment with pixantrone in heavily pretreated patients with DLBCL.

The purpose of the present study is twofold. First, we want to analyse decision-making, preferences and experiences of patients with DLBCL regarding pixantrone treatment. We want to investigate which topics are most relevant for patients at the beginning and during their treatment with pixantrone. Second, we want to evaluate possible changes in the trajectories of the QOL of these patients. The changes in QOL values will be used to compare; patients that respond to treatment with pixantrone versus those who do not respond to treatment; patients that start the drug treatment with pixantrone in an earlier disease setting (third-line treatment) versus those that start the treatment in a later disease setting (fourth line and later); and within patient differences in the QOL values of those that show complete response to pixantrone in the disease progression-free period versus those that show disease progression at the moment of disease progression. Gaining a better understanding of treatment-related decision-making and treatment experiences could facilitate decision-making for future patients and doctors.

### Status of the trial

Due to unexpected delay in patient recruitment, we aim to complete the study in February 2021.

**Correction notice**  This article has been corrected since it first published online. The open access licence type has been amended.

**Acknowledgements**  The authors wish to thank Servier for funding this study.

**Contributors**  SI, CS, JJVB and PL made substantial contributions to conception and design. All authors were involved in drafting the manuscript or revising it critically for important intellectual content. All authors read and approved the final manuscript.

**Funding**  This research grant is funded by Servier Nederland Farma.

**Competing interests**  None declared.

**Patient consent for publication**  Not required.

**Ethics approval**  The Medical Ethical Committee of Erasmus MC, Rotterdam, The Netherlands, has approved this study, registered under MEC-2016-176.

**Provenance and peer review**  Not commissioned; externally peer reviewed.

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
