## [Reviewer comments · BMJ Open]

ARTICLE DETAILS

TITLE (PROVISIONAL)	Patients' decision-making, experiences and preferences regarding Pixantrone treatment in relapsed or refractory diffuse large B-cell lymphoma: study protocol for a longitudinal mixed methods study
AUTHORS	van Hoogdalem, Lothar; Siemes, Claire; Lugtenburg, Pieternella; Busschbach, Jan; Ismail, Sohal

VERSION 1 - REVIEW

REVIEWER	Anne W. Beaven University of North Carolina United States I have received research funding in the past from CTI Biopharma, makers of Pixantrone.
REVIEW RETURNED	01-Oct-2018

GENERAL COMMENTS	This is a very interesting question that may provide information about patient decision making beyond just pixantrone use. I am not surprised that enrollment has been slow due to low patient numbers and the need to perform the first interview and QOL prior to starting pixantrone. If this continues to be a problem then the investigators could consider widening the time interval for obtaining questionnaires so that it could be up to day 7 of cycle 1 of pixantrone. Although this could impact patient responses such as QOL etc. it might still provide meaningful information about decision making.
---

REVIEWER	Paul J. Bröckelmann, MD Department I of Internal Medicine, University Hospital of Cologne, Cologne, Germany
REVIEW RETURNED	06-Nov-2018

GENERAL COMMENTS	Comprehensive ongoing study evaluating the important question of treatment preferences in the setting of a palliative hematologic disease.
--

REVIEWER	Debra Howell University of York. UK
REVIEW RETURNED	05-Dec-2018

GENERAL COMMENTS

Thank you for asking me to review this protocol for a study examining decisions to accept pixantrone treatment in patients with relapsed/refractory DLBCL. The study involves patient interviews, analysed using grounded theory, and a series of questionnaires. It addresses an important issue that is becoming more significant as the range of treatment options expands. However, a number of important flaws have been identified, which should be rectified before publication.

Major issues

PPI: One of my major concerns is that the protocol is not underpinned by any PPI, which weakens the case for the importance of the study. Even if there is no PPI at this stage, there should be a plan for future incorporation of this crucial component. This might include having patients check: study literature (e.g. information sheets) for acceptability to a lay-audience; study methodology for appropriateness, particularly regarding patient contact; and findings, to ensure these resonate with patients.

Pixantrone: More information should be included about this drug, including: how it is administered; and whether it is administered in hospital (thus requiring additional hospital attendance)? Also, what are the alternatives, if any?

Involvement of relatives/carers: It should be made clear if relatives/carers are simply observers, or will be participants in the interviews. It should be made clear if their data will be used in the analysis and if they will be consented.

Death of participants: This is not overtly mentioned in the protocol, although patients included have advanced/refractory disease. HM pathways are known to change abruptly and sudden deaths are common (see work by McCaughan & Howell on end of life care in HMs). It is likely that the sample of 20 will decrease over the time-frame of the study, yet there is nothing in the protocol to indicate this has been considered, or of how it will be managed. Perhaps this should be mentioned in the discussion?

ECOG: How was the decision to include patients with ECOG 2 or better made, rather than also including those with ECOG 3 and 4? This means included patients will be fitter (also younger) than the patient population as a whole. It will impact on the overall transferability of findings. Has this been considered?

Academic level: This might be better phrased as 'education'.

Differences by socio-economic status: This issue is not mentioned but I would have thought this would be identified in the literature as important.

Topic guide: I would expect this to be based on evidence from published literature, as well as the expertise of the study team.

This is particularly important if you have no PPI, so have not based the topic guide on issues determined as important by patients.

Distress/disclosure: What will happen if patients become distressed during the interview? Processes set up to manage this situation should be described, as well as any strategies for disclosure, should this happen.

Number of patients: While 20 patients is an adequate sample for a qualitative study and may lead to data saturation, I am unsure of the value of questionnaires with so few participants, even if these are repeated on multiple occasions. In this context, generalisability should be considered.

Other specific issues

P5, line 137: 'what conditions would patients like to satisfy' - this needs clarifying as I am not sure what it means.

P5 line 139: 'QOL drops down...' - is this an assumption or based on evidence? If the latter, there should be a reference.

	P5, lines 143, 144: 'earlier disease' and 'later disease' - this comes across as subjective. It would be better to use 3rd line and 4th line treatment, as this is more exact. P6, line 159: It would be useful to have a little more information about the hospitals: how many are included; do they provided secondary or tertiary care? P6, line 159: How will you purposively sample? Will it be based on age, sex, SES, ethnicity etc. P6, line 162: How does the researcher find out about which patients are being treated with pixantrone? P6, line 176: 'according to standard clinical practice'; what is this? More details would be helpful. P6, line 201: I'm not sure I would class qualitative interviews as an instrument. P7, lines 209, 210: Interviews will be anonymised (person, place names, staff, family members etc.), might be a better phrase. P7, line 210: Whilst it is acceptable not to return to patients for comments/corrections, the reason for this should be justified. P7, line 223: and compare between patients? P7, line 225: should the title also include the LG QLQ? P8, line 273: This section relates to the questionnaire analysis only. Aspects of the qualitative work should be covered in the section describing these methods. Also (line 279) the methods should be clear enough to 'replicate the study', but findings may differ by patient inclusion, location, hospital provider etc., so generalisability is not an overarching aim. I think this paragraph needs placing elsewhere and amending. I was not able to provide any statistical review. Although largely well written, the paper would benefit from an English language edit.
--	--

REVIEWER	Professor Simon Skene University of Surrey, UK
REVIEW RETURNED	23-Jan-2019

GENERAL COMMENTS	This is an interesting and important study, setting out to understand the experiences of patients regarding important treatment decisions and journeys following repeated treatment failures. The mixed methods are well described, and the authors appear well qualified to undertake the qualitative components. A number of points arise which could be usefully clarified to give greater substance to the paper/protocol.  1. The decision to have no patient or public involvement in such a patient focused study is surprising. Perhaps the authors could explain the rationale for this. Were no patient/advocacy groups consulted over likely burden of questionnaire etc? 2. The procedures to protect patient data could be better described. It is recognised that codes rather than names etc will be used. Will tape recordings be altered to remove identifying information?
---

	3. Is there any steering committee or similar who will protect the interests of patients in the study. Whilst it is recognised that the study is non-interventional, patient contact/involvement is central to the work. 4. The analysis of 'changes' in quality of life is underspecified. To what extent is this analysis limited by the sample size of 20 which is set by the need for data saturation in the qualitative theme analysis? Given the longitudinal nature of the data, other methods (mixed models?) may be useful to better describe the patient trajectories than simple comparisons with baseline. Is a statistician part of the study team? 5. It is not immediately clear what a 'normative' sample for comparison would be for these patients. Perhaps the authors could clarify.
--	---

VERSION 1 – AUTHOR RESPONSE

Reviewer #1 – Anne W. Beaven

1. This is a very interesting question that may provide information about patient decision making beyond just pixantrone use. I am not surprised that enrollment has been slow due to low patient numbers and the need to perform the first interview and QOL prior to starting pixantrone. If this continues to be a problem then the investigators could consider widening the time interval for obtaining questionnaires so that it could be up to day 7 of cycle 1 of pixantrone. Although this could impact patient responses such as QOL etc. it might still provide meaningful information about decision making.

Reaction: We thank the reviewer for her kind and supporting words. If problem with patient enrollment continues, we will certainly consider your advice. As we read, no revisions are suggested by this reviewer. Hence no additional changes have been made in the manuscript.

Reviewer #2 – Paul J. Bröckelmann, MD

2. Comprehensive ongoing study evaluating the important question of treatment preferences in the setting of a palliative hematologic disease.

Reaction: We thank the reviewer for his kind and supporting words. As we read, no revisions are suggested by this reviewer. Hence no additional changes have been made in the manuscript.

Reviewer #3– Debra Howell

3. PPI: One of my major concerns is that the protocol is not underpinned by any PPI, which weakens the case for the importance of the study. Even if there is no PPI at this stage, there should be a plan for future incorporation of this crucial component. This might include having patients check: study literature (e.g. information sheets) for acceptability to a lay-audience; study methodology for appropriateness, particularly regarding patient contact; and findings, to ensure these resonate with patients.

Reaction: We thank the reviewer for her comprehensive review. Page and line referrals refer to the 'track-changes' document. The reviewer makes a very interesting and fair comment. This topic has been discussed with the research team and we are planning to incorporate this component in the study.

4. Pixantrone: More information should be included about this drug, including: how it is administered; and whether it is administered in hospital (thus requiring additional hospital attendance)? Also, what are the alternatives, if any?

Reaction: Administration is intravenously in the hospital. The current treatment situation in patients with relapsed and refractory DLBCL not qualifying for ASCT could be characterized as follows: (1) Beside Pixantrone, there is no approved compound available; (2) there is less consensus about the best way how to treat these patients. This information has been added in the introduction section. [P3 L76 and L82-85]

5. Involvement of relatives/carers: It should be made clear if relatives/carers are simply observers, or will be participants in the interviews. It should be made clear if their data will be used in the analysis and if they will be consented.

Reaction: We agree with you that this information should be stated. Relatives or caregivers are classified as observers. This information is added in the 'design and procedure' section [P5 L188-189].

6. Death of participants: This is not overtly mentioned in the protocol, although patients included have advanced/refractory disease. HM pathways are known to change abruptly and sudden deaths are common (see work by McCaughan & Howell on end of life care in HMs). It is likely that the sample of 20 will decrease over the time-frame of the study, yet there is nothing in the protocol to indicate this has been considered, or of how it will be managed. Perhaps this should be mentioned in the discussion?

Reaction: The reviewer makes a valid comment. The chance of drop-out by death is decreased due to the chosen ECOG PS criteria (see also below). Coping with drop-out is, perhaps poorly, described under the design and procedure section. If patients die before the trial is finished, all collected data will be stored. Qualitative output can still be valuable, even if the patient did not complete the whole trial. As long as our time schedule allow us to, we will strive to make 20 full inclusions. We have updated the design and procedure section [P5 L177 and 179-180].

7. ECOG: How was the decision to include patients with ECOG 2 or better made, rather than also including those with ECOG 3 and 4? This means included patients will be fitter (also younger) than the patient population as a whole. It will impact on the overall transferability of findings. Has this been considered?

Reaction: The reviewer makes a fair point which has been considered before. We have discussed this with our medical staff. There is currently no information on the safety and efficacy of patients with a poor performance status (ECOG > 2) using Pixantrone. Therefore we have decided to exclude patient with ECOG 3 and 4.

8. Academic level: This might be better phrased as 'education'.

Reaction: This word has been corrected according to the advice of the reviewer [P3 L105].

9. Differences by socio-economic status: This issue is not mentioned but I would have thought this would be identified in the literature as important.

Reaction: The reviewer makes a fair point, socioeconomic status is indeed identified as an important factor in literature. In our opinion there is some overlap with 'education', that is why we did not mention it in the first place. After consideration we have updated the manuscript with information about socioeconomic status [P3 L104 and L113-115].

10. Topic guide: I would expect this to be based on evidence from published literature, as well as the expertise of the study team. This is particularly important if you have no PPI, so have not based the topic guide on issues determined as important by patients.

Reaction: We agree that our topic list would benefit from patients' input. During the semi-structured interview, there is enough room for the patient to give their additional input. Moreover, our study follows the guidelines of grounded theory, which means that data generation is a continuous process. Information from a previous interview can be used in a subsequent interview.

11. Distress/disclosure: What will happen if patients become distressed during the interview? Processes set up to manage this situation should be described, as well as any strategies for disclosure, should this happen.

Reaction: The reviewer points out an interesting issue. It is very likely that (such ill) patients become distressed during an interview. The interviewers are well-trained psychologists, they are specifically trained to deal with end of life care. We are unsure of such procedure should be elaborated in the study protocol. A short note regarding this issue is added in the design and procedure section [P5 L189-190 and L191-192].

12. Number of patients: While 20 patients is an adequate sample for a qualitative study and may lead to data saturation, I am unsure of the value of questionnaires with so few participants, even if these are repeated on multiple occasions. In this context, generalisability should be considered.

Reaction: Again, the reviewer makes a fair comment. The output of the interviews is leading and gives us the most valuable information. Generalisability from the questionnaires is indeed limited and we should be careful when interpreting those results.

13. Other specific issues

P5, line 137: 'what conditions would patients like to satisfy' - this needs clarifying as I am not sure what it means.

Reaction: The reviewer makes a fair point. This sentence is deleted after consideration.

P5 line 139: 'QOL drops down...' - is this an assumption or based on evidence? If the latter, there should be a reference.

Reaction: This sentence is based on an assumption.

P5, lines 143, 144: 'earlier disease' and 'later disease' - this comes across as subjective. It would be better to use 3rd line and 4th line treatment, as this is more exact.

Reaction: This sentence is corrected as advised [P4 L148-149].

P6, line 159: It would be useful to have a little more information about the hospitals: how many are included; do they provide secondary or tertiary care?

Reaction: Information regarding this is added in the design and procedure section [P5 L164-166].

P6, line 159: How will you purposively sample? Will it be based on age, sex, SES, ethnicity etc.

Reaction: Purposive sampling in this situation is based on ECOG PS status, age and treatment history. See also our inclusion criteria.

P6, line 162: How does the researcher find out about which patients are being treated with pixantrone?

Reaction: Information regarding this is added in the design and procedure section [P5 L168-169].

P6, line 176: 'according to standard clinical practice'; what is this? More details would be helpful.

Reaction: This means that the chosen questionnaires and interview techniques are common, compared to similar studies.

P6, line 201: I'm not sure I would class qualitative interviews as an instrument.

Reaction: 'Instruments' is replaced by 'Data Collection'. [P6 L213]

P7, lines 209, 210: Interviews will be anonymised (person, place names, staff, family members etc.), might be a better phrase.

Reaction: This phrase is edited according to the advice of the reviewer [P6 L222-223].

P7, line 210: Whilst it is acceptable not to return to patients for comments/corrections, the reason for this should be justified.

Reaction: This phrase is now justified by a reference [P6 L224-226].

P7, line 223: and compare between patients?

Reaction: It is unclear to us what the reviewer means with this comment.

P7, line 225: should the title also include the LG QLQ?

Reaction: Only the EORTC-NHL-HG29 will be administered to patients in this study, therefore we have decided not to mention the LG in the title.

P8, line 273: This section relates to the questionnaire analysis only. Aspects of the qualitative work should be covered in the section describing these methods. Also (line 279) the methods should be clear enough to 'replicate the study', but findings may differ by patient inclusion, location, hospital provider etc., so generalisability is not an overarching aim. I think this paragraph needs placing elsewhere and amending.

Reaction: The reviewer is right that this section contains information that belongs somewhere else. Information regarding qualitative data has been replaced under 'sample size calculation' [P8 L288-295].

Reviewer #4 – Professor Simon Skene

14. This is an interesting and important study, setting out to understand the experiences of patients regarding important treatment decisions and journeys following repeated treatment failures. The mixed methods are well described, and the authors appear well qualified to undertake the qualitative components. A number of points arise which could be usefully clarified to give greater substance to the paper/protocol.

The decision to have no patient or public involvement in such a patient focused study is surprising. Perhaps the authors could explain the rationale for this. Were no patient/advocacy groups consulted over likely burden of questionnaire etc?

Reaction: We thank the reviewer for his kind and supporting words. This topic has been discussed with the research team and we are planning to incorporate this component in the study.

15. The procedures to protect patient data could be better described. It is recognised that codes rather than names etc will be used. Will tape recordings be altered to remove identifying information?

Reaction: The reviewer makes a fair point. Additional information regarding this issue is added in the manuscript [P6 L221-222].

16. Is there any steering committee or similar who will protect the interests of patients in the study. Whilst it is recognized that the study is non-interventional, patient contact/involvement is central to the work.

Reaction: Every participant receives a patient information leaflet which contains the contact details of 1) the principal investigator of the study, 2) an independent physician and 3) an independent complaints committee of our Medical Centre.

17. The analysis of 'changes' in quality of life is underspecified. To what extent is this analysis limited by the sample size of 20 which is set by the need for data saturation in the qualitative theme analysis? Given the longitudinal nature of the data, other methods (mixed models?) may be useful to better describe the patient trajectories than simple comparisons with baseline. Is a statistician part of the study team?

Reaction: The reviewer makes a fair comment. The output of the interviews is leading and gives us the most valuable information. Generalisability from the questionnaires is indeed limited and we should be careful when interpreting those results. A statistician is working on our department (dr. R. Timman), he can be counselled any time when we have statistical issues.

18. It is not immediately clear what a 'normative' sample for comparison would be for these patients. Perhaps the authors could clarify.

Reaction: We aim to obtain normative data from the EORTC-QLQ-C30 and EQ-5D, please see also references below.

van de Poll-Franse LV, Mols F, Gundy CM, Creutzberg CL, Nout RA, Verdonck-de Leeuw IM et al. Normative data for the EORTC QLQ-C30 and EORTC-sexuality items in the general Dutch population. Eur J Cancer. 2011;47(5):667-75.

Konig HH, Bernert S, Angermeyer MC, Matschinger H, Martinez M, Vilagut G et al. Comparison of population health status in six european countries: results of a representative survey using the EQ-5D questionnaire. Med Care. 2009;47(2):255-61.

VERSION 2 – REVIEW

REVIEWER	Debra Howell University of York, UK
REVIEW RETURNED	19-Feb-2019

GENERAL COMMENTS	The paper is much improved. I am still concerned about the lack of PPI and think the plan to include this, as mentioned by the authors, should be incorporated into the appropriate section, rather than leaving it blank.
--

REVIEWER	Professor Simon Skene University of Surrey, UK
REVIEW RETURNED	06-Feb-2019

GENERAL COMMENTS	The authors have adequately addressed my concerns. I would have preferred some of the response to be incorporated in the article (ie commitment to enhanced PPI, limitations of the quantitative analysis), but am otherwise happy with the revised document.
---

VERSION 2 – AUTHOR RESPONSE

Reviewer #3 and #4 – Debra Howell & Simon Skene

#3: The paper is much improved. I am still concerned about the lack of PPI and think the plan to include this, as mentioned by the authors, should be incorporated into the appropriate section, rather than leaving it blank.

#4: The authors have adequately addressed my concerns. I would have preferred some of the response to be incorporated in the article (ie commitment to enhanced PPI, limitations of the quantitative analysis), but am otherwise happy with the revised document.

Reactions: We thank reviewer #4 for accepting the paper in the current form. We thank both reviewers for acknowledging the improvements. However, we respect both reviewers' directions towards PPI and limitations of the study and will therefore address these items in the paper. We had already considered patient involvement but decided not to do so given the disease burden and the very limited number of eligible patients in this group. To see if we could add some PPI elements to study (already commenced) we have taken notice of the PPI literature (Gray-Burrows KA, Willis TA, Foy R, et al. Role of patient and public involvement in implementation research: a consensus study. *BMJ Qual Saf* 2018;27:858-864.) and decided to allow patient participation in conducting the study and sharing/using research knowledge (see marked copy; P9, line 366 – 373).